# Optically Transparent Bamboo: Preparation, Properties, and Applications

**DOI:** 10.3390/polym14163234

**Published:** 2022-08-09

**Authors:** Xuelian Li, Weizhong Zhang, Jingpeng Li, Xiaoyan Li, Neng Li, Zhenhua Zhang, Dapeng Zhang, Fei Rao, Yuhe Chen

**Affiliations:** 1School of Art and Design, Zhejiang Sci-Tech University, Hangzhou 310018, China; 2Key Laboratory of High Efficient Processing of Bamboo of Zhejiang Province, China National Bamboo Research Center, Hangzhou 310012, China

**Keywords:** transparent bamboo, optical properties, mechanical properties, functionalization and application

## Abstract

The enormous pressures of energy consumption and the severe pollution produced by non-renewable resources have prompted researchers to develop various environmentally friendly energy-saving materials. Transparent bamboo represents an emerging result of biomass material research that has been identified and studied for its many advantages, including light weight, excellent light transmittance, environmental sustainability, superior mechanical properties, and low thermal conductivity. The present review summarizes methods for preparing transparent bamboo, including delignification and resin impregnation. Next, transparent bamboo performance is quantified in terms of optical, mechanical, and thermal conductivity characteristics and compared with other conventional and emerging synthetic materials. Potential applications of transparent bamboo are then discussed using various functionalizations achieved through doping nanomaterials or modified resins to realize advanced energy-efficient building materials, decorative elements, and optoelectronic devices. Finally, challenges associated with the preparation, performance improvement, and production scaling of transparent bamboo are summarized, suggesting opportunities for the future development of this novel, bio-based, and advanced material.

## 1. Introduction

As environmental concerns and the desire to establish a sustainable civilization become more urgent, bamboo has been identified as a potential replacement for materials based on non-renewable resources. There are about 1500 species of bamboo and 36 million hectares of bamboo planting area widely distributed across America, Asia, and Africa [1]. Indeed, bamboo is an important forest resource, having a higher yield, more rapid growth rate, and better mechanical properties than wood, as well as a high aspect ratio and excellent biodegradability. In terms of growth rate, bamboo has a short growth cycle of 3–5 years, whereas wood has a growth cycle of 20–60 years. Furthermore, single bamboo fiber has average tensile strength and modulus of 1.6 GPa and 33 GPa, respectively [2], which is significantly higher than other known natural fibers, such as cotton, coir, henequen, and ramie [3]. Therefore, bamboo has been widely used to fabricate various structural composites, including bamboo scrimber composites [4,5], laminated bamboo lumber [6,7], and bamboo-fiber-reinforced epoxy composites [8].

Recently, interest has arisen in the use of transparent wood (TW) to reduce energy consumption in buildings by capitalizing on its transparency, high mechanical properties, and excellent thermal insulation performance [9,10]. Fabricated by extracting lignin and filling the remaining wood template with a resin, TW exhibits excellent optical transmittance across a wide range of wavelengths [11]. Typically, TW is formed by combining the skeleton structure of wood with transparent organic resins such as an epoxy polymer (EP) or polymethyl methacrylate (PMMA) [12,13]. Using this approach, TW inherits the excellent characteristics of native wood, including a high modulus, low density, high strength, and high toughness [14]; similarly, its transparency depends on the composition and structure of the specific wood employed [15].

As a natural composite material, wood consists of cellulose, hemicellulose, and lignin, which are arranged in a multi-layered and hierarchical structure. Wood is naturally opaque and always has a certain color owing to two aspects. On the one hand, the chromophoric groups in lignin and other substances have a strong ability to absorb visible light at wavelengths of 380–780 nm [16,17], accounting for 80–95% of the total light absorption of wood and constituting the fundamental reason for its color [18]. On the other hand, many pores in the wood are primarily filled with air and water. This mesoporous structure scatters a great deal of light in the visible wavelength range [19,20]. Moreover, the different components in the wood structure create interfaces between materials with different refractive indexes that induce light refraction, scattering, and reflection when light propagates across them [21]. Therefore, removing the colored substances in wood and filling the cavities with a resin providing a refractive index approaching cellulose can make wood transparent.

Bamboo and wood share many characteristics: they have a highly aligned hierarchy and contain similar primary components [22], including cellulose, hemicellulose, and lignin. Therefore, bamboo also has the potential to be prepared as a transparent material. As mentioned above, bamboo has many advantages compared with wood. However, bamboo’s high density and low porosity present notable challenges in preparing transparent bamboo (TB) [23]. For example, the high density of bamboo reduces its permeability during treatment; the density of mature bamboo (~0.65 g/cm^3^) is much higher than that of the low-density wood species such as balsa, basswood, and poplar (normally ~0.1–0.4 g/cm^3^), which are typically used to produce TW [23].

Furthermore, the poor permeability of bamboo increases the time and quantity of chemicals required to remove the lignin. Poor permeability also affects the resin filling rate, potentially reducing TB’s light transmittance and mechanical properties [24]. As a result, recent studies have shown that applying the TW preparation process directly to the preparation of TB results in light transmittance of less than 10% [25]. The preparation of transparent bamboo composites (TBCs), which comprise TB combined with other materials to improve performance, faces many of the same challenges.

Therefore, this review summarizes recent advances in the production of functional TB to identify potential directions for future development. First, several simple and efficient TB preparation strategies are introduced based on delignification/lignin modification and resin impregnation. The optical and mechanical properties of TB prepared using different methods are then compared, and finally, various potential applications of TB in energy-efficient buildings, decorative elements, and optoelectronic devices are discussed.

## 2. Basics of Light–Bamboo Interaction

As light travels through the air and interacts with solids, its propagation can continue in the forward direction when refracted and/or absorbed, or it can be reflected backward at the air–solid interface (Figure 1a). In order to effectively discuss the optical properties of transparent materials, it is necessary to define the corresponding terms [26]. The total optical transmittance of an object (often referred to simply as its transmittance) is the ratio of the transmitted light intensity (including the intensities of the directly transmitted light *I*_T,direct_ and diffused transmitted light *I*_T,diffuse_) to the incident light intensity *I*_I0_; the total transmittance is, therefore (*I*_T,direct_ +*I*_T,diffuse_)/*I*_I0_. The optical haze is the ratio of *I*_T,diffuse_ to the total transmitted light and is therefore defined as *I*_T,diffuse_/(*I*_T,direct_ +*I*_T,diffuse_), as shown in Figure 1a. Owing to the difference between refractive indexes, when light passes through an object, it will be refracted at an angle obeying Snell’s law, expressed as *n*_1_sin*θ*_1_ = *n*_2_sin*θ*_2_, where *n* is the refractive index of each material and *θ* is the incident light angle in that material. Multiple factors can influence an object’s optical transmittance and haze, including its surface roughness, thickness, refractive index, pore size distribution, porosity, etc. [27,28,29]. For two-phase materials, such as microscale composites, the higher the refractive index (RI) ratio between the two media, the stronger the light scattering, which corresponds to a larger proportion of reflected light and thus a lower transmittance. The attenuation of light occurs when light is transformed into other types of energy, such as heat. The more solid–solid interfaces in a composite or the greater its thickness, the lighter attenuation that occurs, thus reducing transmittance. Thus, a transparent composite material can be realized by providing a low RI ratio between phases, low light attenuation, and lesser thickness, resulting in a higher transmittance.

Bamboo is opaque because of its optically heterogeneous nature, a result of its microscale porous structure, different chemical components with different RIs in the cell walls, and contents of strongly light-absorbing chemical entities. Figure 1b illustrates the microstructure of bamboo. When light interacts with bamboo, a combination of scattering/reflection, refraction, transmission, and absorption occurs. Light scattering takes place at all interfaces between the cell walls (which have an RI of ~1.56) and air (which have an RI of 1.0). Inside the cell walls, the mismatch between the RIs of the main chemical components of lignin (1.61), cellulose (1.53), and hemicellulose (1.53) leads to additional light scattering [30].

Furthermore, lignin exhibits particularly strong light absorption among these components, accounting for 80–95% of all light absorbed by bamboo [18]. The details of this interaction depend on the light’s wavelength and the bamboo’s properties, such as its density, chemical compositions, and fiber direction. Generally, to make bamboo transparent, light absorption by chemical entities and light scattering at the air/cell wall interfaces and inside the cell walls must be reduced or eliminated.

**Figure 1 polymers-14-03234-f001:**
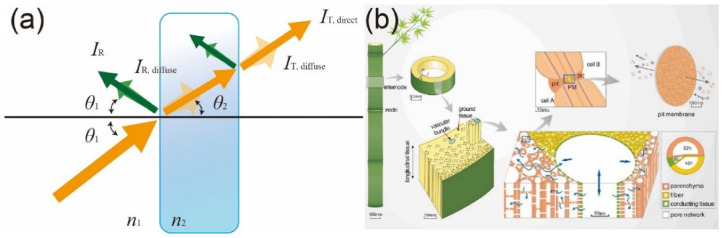
(**a**) Sketch illustrating light propagation through a solid object in a medium (i.e., air), where *I*_IO_ is the incident light intensity, *I*_T,direct_ is the intensity of directly transmitted light, *I*_T,diffuse_ is the intensity of diffusely transmitted light, *I*_R_ is the intensity of reflected light, *I*_R,diffuse_ is the intensity of diffusely reflected light, *θ*_1_ is the incident angle in the medium, *θ*_2_ is the refracted angle in the solid, and *n*_1_ and *n*_2_, respectively, represent the refractive indices of the medium and solid. (**b**) Schematic of the hierarchical pore network in bamboo, where PM indicates the pit membrane [31]. ((**a**) Adapted from Li et al. [32] Copyright 2017 the Royal Society; (**b**) Reproduced with permission from Liu et al. [31]. Copyright 2021. Elsevier).

## 3. Preparation of TB

The preparation of TB can be divided into two steps. First, the color-producing compounds in bamboo are removed or modified (decolorization treatment); second, the bamboo is impregnated with a RI-matched resin. Removing or modifying the lignin and chromophoric groups of bamboo is a particularly crucial step. However, the delignified or lignin-modified bamboo is still opaque as the RIs of the substrate and air still do not match. Therefore, it is essential to impregnate the bamboo with a transparent resin providing matching RI to make the bamboo transparent.

### 3.1. Preparation of the Bamboo Template

The first key step in the preparation of TB is to remove or chemically modify the chromogenic substances in the bamboo (primarily lignin) to achieve a decolorized template. Decolorization is commonly achieved in wood and bamboo using acid delignification, alkali delignification, lignin modification, or enzyme delignification methods.

#### 3.1.1. Acid Delignification Method

The preparation of delignified bamboo templates using the relatively simple acid delignification method was demonstrated by Wu et al. [25], who employed a certain concentration of sodium chlorite (NaClO_2_) mixed with water and acetic acid (CH_3_COOH) to treat bamboo at a pH value of 4.6 and a temperature of 80–90 °C until it turned white. The time required for delignification differed according to the size of the bamboo sample. Under this method, the NaClO_2_ forms unstable chlorous acid in an acidic environment; this chlorous acid decomposes into Cl_2_, ClO_2_, and H_2_O, which interact with the benzene ring structures in the conifer aldehyde and aromatic ketone in the lignin through an oxidative ring-opening reaction to form an acidic group, causing the material to degrade and dissolve in water. Furthermore, the hypochlorous acid produced by the reaction of Cl_2_ with water is also a strong oxidant that reacts with lignin to finally produce o-quinone, small molecules (e.g., carboxylic acid) and corresponding alcohols. Lignin macromolecules are broken and dissolved through these reactions, thereby achieving the purpose of decolorization. When sodium hypochlorite (NaClO) is used to remove lignin instead, the main reactions include chlorination and oxidation, and the main reaction objects are the benzoquinone structure of the lignin benzene ring and the conjugated double bond of the side chain. These reactions are able to generate small molecules (such as CO_2_ and carboxylic acid) and thereby remove lignin and other coloring substances from the sample [33,34].

In addition, the use of 1% sodium hydroxide (NaOH) to pretreat natural bamboo prior to acid delignification has been observed to help prepare TB with high optical transmittance [23]. Figure 2a summarizes the associated bamboo template preparation stages. The transmittance of TB prepared using NaOH and acid delignification was found to be ~8% lower than that of TW, but its cellulose volume fraction was ~400% higher, resulting in a tensile strength of 92 MPa, roughly twice that of TW. Furthermore, the developed TB exhibited a low heat conductivity of 0.203 Wm^−1^ K^−1^, indicating a reduction of ~10% and ~80% compared to TW and traditional glass, respectively. Indeed, the pretreatment of bamboo with NaOH results in a more microporous cell wall structure, primarily owing to the partial removal of extract, hemicellulose, and lignin from the cell walls to form small cavities. These cavities’ existence increases the cell wall’s contraction and thereby decreases its thickness, and realizing a porous structure more conducive to the infiltration of lignin-removal chemicals.

#### 3.1.2. Alkali Delignification Method

Researchers have also prepared TW templates by treating wood samples with a mixed solution of NaOH and sodium sulfite (Na_2_SO_3_) at 100 °C for 12 h, as SO_3_^2−^ can sulfonate lignin in alkaline conditions [36,37]. The purpose of this lignosulfonalization is to introduce sulfonic acid groups into the side chain of the benzene rings in lignin. The reaction products are not only soluble in water but can also break the bonds of various ethers. Considering that the quinone structure formed during alkaline sulfite treatment will darken the color of the sample, further bleaching with hydrogen peroxide (H_2_O_2_) is required [15]. Notably, H_2_O_2_ bleaching is also a delignification process. First, H_2_O_2_ is dissociated in an alkaline solution to form hydrogen peroxide anion (HOO^−^), which can decompose the quinone structure and even degrade it into small molecule esters. It should be noted that the bleaching effect of HOO^−^ is closely related to the pH of the solution. When the pH increases, the concentration of HOO^−^ increases, leading to an improved bleaching effect. However, when the pH exceeds 10.5, HOO^−^ is easily decomposed into O_2_, and the bleaching effect will deteriorate [15]. Critically, the alkali delignification method is complex and can readily cause sample deformation [38].

#### 3.1.3. Lignin Modification (Lignin Retaining) Method

The lignin modification method has been used to prepare bamboo templates for TB by retaining the lignin while modifying the chromophoric groups within, as shown in Figure 2b. Because the lignin composition is retained, the mechanical strength of the resulting TB is typically higher than that of TB prepared by removing the lignin [35]. Moreover, the lignin modification method also avoids wasting the lignin components. For example, Wang et al. [35] removed the light-absorbing chromospheres (aromatic ketone, coniferaldehyde, and orthoquinone) of lignin using an alkali H_2_O_2_ treatment while retaining the aromatic skeleton lignin structure (Figure 2b). To do so, sodium silicate (3.0 wt%), sodium hydroxide (3.0 wt%), diethylenetriamine pentaacetate (0.1 wt%), magnesium sulfate (0.1 wt%), and H_2_O_2_ (4.0 wt%) were dissolved in deionized water to prepare the lignin modification solution in which the bamboo samples were immersed at 70 °C until they turned completely white. The lignin-modified bamboo was then thoroughly rinsed three times with boiled deionized water to remove any residual chemicals. Finally, the resulting bamboo templates were solvent exchanged using acetone and ethanol. This lignin modification method was confirmed to selectively remove or react the chromogenic groups in the bamboo while retaining most of the lignin. The resulting high-lignin-content TB had a transmittance of 87% and haze of 90%.

#### 3.1.4. Biological Enzyme Delignification

Delignification using the biological enzyme method represents an environmentally friendly process, as the need for harmful chemicals is minimized. Jichun and Yan [39] used biological enzymes to degrade lignin and thereby achieve the decolorization of wood by employing the following procedure. The dried wood samples, pure water, biological enzyme (synthetic laccase/xylanase system at a dosage of 10 IU/g), and glacial acetic acid were combined in a 1:30–40 mass ratio of sample to water, and the pH was adjusted to 3–5 by adding trace quantities of hydrogen peroxide (up to 4% of the sample mass). A treatment temperature of 35–50 °C was then applied for 1–2 h, after which the samples were washed with deionized water. Next, the samples were extracted using 30 wt% dioxygen water and 25 wt% ammonia water at a volume ratio of 10:1. The extracted samples were then washed with deionized water and dehydrated by ultrasonic extraction to obtain the TW templates.

## 4. Properties of TB

### 4.1. Optical Properties

Ongoing research has produced TB with excellent optical transmittance and high haze (Table 1). There are three main interactions with light to consider when evaluating the optical properties of TB: (1) reflection at the outer gas/TB interface, (2) scattering in the form of reflection and refraction, and (3) absorption inside the TB. In TB, light scattering mainly occurs at the interface between the bamboo tissue and the polymer. The lower the difference between the RIs of the bamboo template and polymer, the less scattering will occur at their interface. High haze is primarily a result of collective scattering inside the composite material. Finally, light absorption is primarily caused by the presence of lignin in the bamboo. Delignification and lignin modification have accordingly been applied, as discussed in Section 3, to remove the components containing chromophores and thereby minimize light absorption.

As shown in Table 1, the optical properties of TB prepared using the same resin differ substantially according to the preparation method. As discussed in Section 1, the light transmittance obtained by applying the TW preparation process directly to the preparation of TB was not ideal [25]. However, the light transmittance of TB was effectively increased to 80% by pretreating with 1% NaOH before acid-chlorite delignification, as this increased the contraction of the cell wall and decreased its thickness through the partial removal of extract, lignin, and hemicellulose [23]. In addition, the lignin modification method produced TB with excellent light transmittance as high as 87% by removing the light-absorbing chromospheres while preserving the aromatic skeletal lignin structure [35].

The most frequently used resin for the preparation of TB is shown in Table 1 to be EP, primarily owing to the similarity of its RI to that of the bamboo template. This reduces the degree of scattering to obtain better optical performance. Factors such as the fiber volume fraction and thickness of the bamboo template will also affect the optical properties of TB. The volume fraction of bamboo fibers naturally increases from the inner side to the outer side of the bamboo culm; TB with the desired fiber volume fraction can be obtained for a given thickness by compressing the bamboo template prior to resin impregnation.

As shown in Figure 3a, the transmittance of TB generally decreases with increasing fiber volume fraction, primarily because a higher fiber volume fraction hinders the infiltration of the lignin-modification chemicals into the bamboo, thereby leaving more chromospheres in the lignin [35]. Figure 3a also shows the transmittance of the inner layer with a low fiber volume fraction is higher than that of the outer layer with high fiber volume fraction. Furthermore, as the bamboo thickness increases, the transmittance decreases and haze increases, primarily owing to the longer light pathway, which corresponds to an increasing degree of light attenuation and an increasing number of polymer/bamboo interfaces, leading to more light scattering. Indeed, as reported in Figure 3b, as the TB thickness increased from 0.3 to 2.9 mm, the transmittance decreased from 92.4% to 23.7%, and the haze increased from 43.5% to 82.95% [24]. Figure 3b also shows that the transmittance of multi-layer TB is far higher than that of single-layer TB with the same thickness because multi-layer TB enables more uniform epoxy resin impregnation through the laminations. For example, the transmittance of 1.2 mm thick multi-layer TB was found to be 78.6, whereas that of 1.2 mm thick single-layer TB was 10.4 [24].

In addition, TB also demonstrates an anisotropic optical property attributed to cellulose orientation-induced birefringence (△n ≈ 0.074–0.08) and the anisotropic structure of natural bamboo [40,41]. Indeed, the haze of TBC made from TB is higher in the radial direction than in the longitudinal direction (Figure 3c) because TBC has a lower density of polymer/cellulose interfaces in the longitudinal direction owing to the hollow cylindrical shape of bamboo cells [35]. On the one hand, the light scattering is highly anisotropic when the light is perpendicular to the longitudinal direction of the TBC; on the other hand, the light scattering is almost isotropic when the light is oriented along the longitudinal direction of the TBC (Figure 3d) [35]. Thus, different relationships between the orientation of the bamboo fibers and the direction of light (perpendicular and parallel) will result in different scattering patterns.

### 4.2. Mechanical Properties

The mechanical properties of TB are affected by the preparation method, bamboo properties such as cellulose content, cell structure morphology, and density; bamboo structural anisotropy; and the interfacial bond between the bamboo template and the impregnating resin. In addition, a summary including mechanical and physical properties of natural fibres is shown in Table 2.

As shown in Table 1, the mechanical properties of TB produced using the same resin exhibit substantial differences based on the applied preparation method. Notably, as the lignin modification method removes the light-absorbing chromophore groups without entirely destroying the aromatic structure of lignin, the mechanical properties of the resulting TB are much better. As a result, the tensile strength of TB obtained by lignin modification can be as high as 118 MPa [35].

The mechanical properties of TB are strongly dependent on the mechanical properties of the bamboo template. Density, which mainly depends on the fiber diameter, fiber content, and cell wall thickness, exerts a considerable influence on the mechanical properties of bamboo. The bamboo fiber density increases with increasing fiber content. As mentioned in Section 4.1, the bamboo fiber volume fraction increases from the inner side to the outer side of the bamboo culm, resulting in different bamboo layer densities according to their original locations within the stalk. Thus, TB made from an outer bamboo layer is often stronger than TB made from an inner layer, as shown in Table 1. Note that Table 1 also indicates that the tensile strength of multi-layer TB is higher than that of single-layer TB with the same thickness. In the beginning, the tensile strength of 1.2 mm thick multi-layer TB was found to be 61.89 MPa, whereas that of 1.2 mm thick single-layer TB was found to be 61.15 MPa [24]. As the TB thickness increases, the difference in tensile strength grows larger. This result is related to interfacial compatibility. As mentioned in Section 4.1, multi-layer TB enables more uniform resin impregnation through the laminations, whereas it is difficult for the resin to infiltrate into the bamboo cells of single-layer TB. Therefore, the tensile strength of multi-layer TB is greater, making it more suitable as a structural material in electronics, household, and construction applications. Furthermore, the tensile strength of TB first increases and then decreases with increasing bamboo template thickness. A thicker bamboo template makes it difficult to completely remove lignin but and hinders the ability of the resin to infiltrate into the template. In contrast, a relatively thinner bamboo template exhibits more extensive lignin removal and can easily be impregnated with resin. The properties of TB can thus be tailored by compression of the bamboo template to provide different densities and thicknesses. Furthermore, owing to the anisotropic structure of bamboo, which has no radial cell elements, the tensile strength of TB is much higher in the longitudinal direction (about 118 MPa) than in the radial direction (about 30 MPa) [35].

### 4.3. Thermal Conductivity

Excellent thermal insulation performance with low heat conductivity is critical to realizing energy-efficient building materials [48,49]. The thermal conductivity of TB (0.203 W m^−1^ K^−1^) is accordingly compared with that for common glass (0.974 W m^−1^ K^−1^), TW (0.225 W m^−1^ K^−1^) [23], and other materials in Figure 4a. Notably, the figure indicates that TB has a lower thermal conductivity than TW. The low thermal conductivities of both TB and TW can be attributed to the interfacial heat resistance and the phonon dispersion between the air and cell walls [50]. Thermal conduction in TB is also anisotropic, like its optical and mechanical properties. Indeed, the thermal conductivities of TB in the longitudinal and radial directions were found to be about 0.44 and 0.33 W m^−1^ K^−1^ (Figure 4b), respectively [35]. This indicates that the thermal energy tends to spread more in the direction parallel to the direction of bamboo growth owing to the orientation of the cellulose nanofibers [51]. Because of its low thermal conductivity yet relatively high light transmittance, TB represents a promising window material that can prevent heat dissipation and thus reduce the energy consumption of buildings.

## 5. Functionalized TB and its Potential Applications

Bamboo has considerable multifunctional potential owing to its multi-scale pores and composition of lignin, hemicellulose, and cellulose. Typical modification strategies to improve the functionality of TB include cell wall modification, cell wall/lumen interface modification, and lumen filling, as shown in Figure 5. Notably, the bamboo template used to produce TB has been shown to demonstrate higher porosity than the original bamboo, potentially facilitating additional modification [23]. One method for TB modification is to fill the lumen space with polymer liquids decorated with functional nanoparticles, followed by polymerization and curing. Another TB modification method is to first modify the bamboo template, then apply monomer impregnation and polymerization. Taking TW as an example, successful modifications have realized ultraviolet (UV)-stabilized wood [54,55], magnetic wood [56,57], conductive wood [58,59,60], and stimuli-responsive wood [61]. Bamboo also has the potential to be prepared with the same functionalities owing to its similar structure and composition. Such functionalizations can enable the wide application of TB in fields such as energy-saving windows, decorative elements, and optoelectronic devices.

### 5.1. Energy-Saving Windows

Reducing building energy consumption has been the focus of much research. Windows are essential building components considering the requirements of human visual comfort and health; however, they are also the least energy-efficient elements of a building since energy is always transferred through a window in the direction opposite to that desired. In summer, when the outdoor temperature is higher than the indoor temperature, heat is transferred to the indoor environment through the windows; in winter, when the outdoor temperature is lower than the indoor temperature, the heat inside is released into the outdoor environment. Both phenomena increase the energy consumption required to maintain a comfortable indoor environment. Therefore, smart windows and low emissivity (low-E) glass have been designed to alleviate these problems [63,64,65,66]. However, there remain several issues with conventional low-E glass. Most importantly, low-E coatings are produced using either physical vapor deposition (PVD) or chemical vapor deposition (CVD), which require expensive, high-vacuum setups [67,68,69].

To address these problems, two kinds of TWCs (transparent wood composites) with anisotropic structures were developed to serve as energy-saving windows. These TWCs employed the infiltration of an EP resin dispersion containing 0.04 wt% tungsten-doped vanadium dioxide nanoparticles (W/VO_2_ NPs) into a delignified wood template in the longitudinal or radial directions (W/VO_2_-TPW-R or W/VO_2_-TPW-L, respectively), followed by subsequent polymerization, as shown in Figure 6a [70]. Both TWCs exhibited excellent temperature regulation when used as windows: after 5 min of solar radiation, the temperature of a simulated room with a glass window rose sharply from 21.0 to 60.7 °C, whereas that of a room with an EP-resin impregnated TW window increased from 21.0 to 47.2 °C, that of a room with the W/VO_2_-TPW-R window only increased from 21.0 to 40.6 °C, and that of a room with the W/VO_2_-TPW-L window only increased from 21.0 to 39.1 °C. Thus, W/VO_2_-TPW-R/L provided the most effective thermal insulation among the investigated window materials. The superior effectiveness of this material can be attributed to the large quantity of heat reflected by the VO_2_ NPs. As a result, the indoor temperature increased at a significantly slower rate than when using ordinary glass panels. This novel TWC combining a low thermal conductivity TW template with thermochromic VO_2_ NPs provided a potential solution for replacing expensive, heavy, high thermal conductivity and infrared transparent glass.

Similarly, energy-saving windows made of TB-based materials combining a low-E coating with radiative cooling (RC) have been reported. A material capable of RC spontaneously cools its surface by reflecting sunlight and radiating long wavelength infrared (LWIR) to the cold outer space. This technique can deliver effective cooling power through highly emissive (ε_LWIR_) materials, though applications of related technology to windows have not been extensively explored [71]. To achieve the maximum cooling effect, the side of the window facing the outside must have a high ε_LWIR_ to promote RC, while the other side must have a low ε_LWIR_ to prevent heat exchange [72,73,74]. However, though the RC technique can reduce the energy consumption required for cooling in summer, it also increases the heating burden in winter owing to its heat-releasing characteristic. Therefore, manufacturing a TBC with enhanced RC for application in energy-saving windows could reduce the energy consumption when cooling is required.

Cellulose-based glass panes with high optical transparency were accordingly obtained by delignification of bamboo slices followed by epoxy infiltration, then depositing silver nanowire on one face to realize RC cellulose glass [70]. A low indoors-facing ε_LWIR_ of 0.3 was thereby achieved, together with a large ε_LWIR_ difference of 0.65 between the two faces. A schematic demonstration of the working mechanism of this RC cellulose glass is shown in Figure 6b. The RC performance was considerably enhanced by the suitable difference between the two faces ε_LWIR_. Critically, compared to conventional low-E glass, RC cellulose can be produced using a solution-based process rather than complex PVD or CVD processes. Thus, low-cost and eco-friendly, energy-saving windows derived from bamboo demonstrate the considerable potential of TB functionalization in architecture.

**Figure 6 polymers-14-03234-f006:**
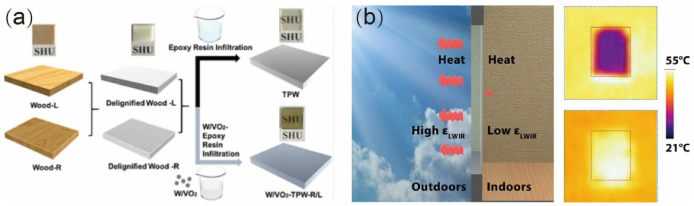
Functional transparent composite materials for energy-saving windows. (**a**) Schematic description of the synthesis route of tungsten-doped vanadium dioxide nanoparticles impregnated into a delignified wood template in the longitudinal or radial directions (W/VO_2_-TPW-R/L) [70]. (**b**) Schematic demonstration of the working mechanism of radiative cooling (RC) cellulose glass [75]. ((**a**) Reproduced with permission from Zhang et al. [70]. Copyright 2020 ACS Publications; (**b**) Reproduced with permission from Zhou et al. [75]. Copyright 2021 ACS Publications).

### 5.2. Decorative Applications

The most obvious advantage of TB is that it conforms to the concept of sustainable and environmentally friendly development. Compared with glass, plastic, and other transparent household materials, TB is derived from renewable biomass materials and is produced using a low-cost, non-toxic process. Furthermore, TB has high light transmittance (over 80%) and haze (over 80%) as well as excellent mechanical properties with a higher tensile strength than wood. Though the tensile strength of bamboo is weakened when its internal fibers are destroyed during the delignification process, the tensile strength of the TB is increased after resin impregnation. Therefore, TB is not only useful for household decoration but also in the load-bearing components of household products.

Indeed, TB can serve as a replacement for transparent materials such as glass or plastic in many decorative applications. Furthermore, the application of TB in household products can increase the sense of transparency and mystery [76]. It has been found that overlapping certain thicknesses of TB can produce different effects of transmittance and haze, providing the possibility of forming different visual effects through weaving [77]. For example, a screen woven from TB is shown in Figure 7a. This screen design not only combines TB and wood but also highlights the transparency and haze of TB and reflects aspects of traditional Chinese culture using new materials. Lamp design represents another crucial part of the home decoration industry and should consider the application of new materials and technologies. Indeed, TB can address many deficiencies in the field of lamp design in China, including a serious homogeneity among products, lack of innovation, and design obsolescence. For example, the upper and lower lampshades shown in Figure 7b are made of TB, softening the light. The advantages of TB can thus be fully realized in the design of household decorative products while collocating it in mixed applications with other materials to achieve multiple styles. It is expected that designers will use the patterns present in TB or dope TB with multifarious guest materials to develop next-generation decorative materials and even furniture that possess high transparency, haze, and fascinating patterns to serve various interesting functions.

### 5.3. Optoelectronic Devices

Transparent biomass materials offer a promising base for optoelectronic devices being developed for a wide range of applications such as mobile phone cameras, new energy conversion systems, and medical testing. Biomass-based transparent materials also have potential as structural materials for photovoltaic equipment such as solar cells [78,79,80]. Indeed, solar cell efficiency has been improved by 18.02% by placing TW on the top surface (Figure 8a) [78], as its high haze leads to large scattering angles that increase the light path length within the solar cell below. Magnetic TW has been obtained by adding magnetic nanoparticles during preparation [81,82]. Furthermore, TW chips treated with luminescent γ-Fe2O3@YVO4:Eu3+ nanoparticles have been shown to emit light under UV excitation (Figure 8b). Such TW functionality has a wide range of potential applications in luminous magnetic switches, lighting equipment, and anti-counterfeiting equipment.

In addition, a novel, stretchable, transparent, and electroconductive wood material has been fabricated by filling the channels and pores of delignified wood with poly (PDES (polymerizable deep eutectic solvents)) via in situ photopolymerization [83]. The combination of poly (PDES) with the wood matrix imbued the material with excellent transmittance (about 90%), stretchability (up to 80% strain), and superior electrical conductivity (up to 0.16 S m^−^^1^).

A transparent conductive wood (TCW) slice can clearly show the parts beneath or make an embedded LED glow (Figure 8c), demonstrating its excellent transparent and conductive characteristics. Furthermore, its unique and outstanding performance characteristics mean that TCW exhibits excellent functional abilities to sense external stimuli, especially in terms of strain and touch. This allows TCW-based sensors to detect subtle human bending release activities and other weak pressures (Figure 8d).

Moreover, a completely wood-based flexible electronic circuit has been reported in which the substrate of the circuit—a strong, flexible, and transparent wood film (TWF)—was printed with a sustainable, bio-based lignin-derived carbon nanofiber conductive ink (Figure 8e) [84]. Notably, the TWF fabrication process maintained the original alignment of the cellulose nanofibers and promoted their combination in the process of removing lignin and hemicellulose. The Young’s modulus and tensile strength of this TWF were determined to be 49.9 GPa and 469.9 MPa, respectively, which are greater than those of most natural fibers, plastics, polymers, and metal or engineered alloys, as shown in Figure 8f [84]. The combination of TWF and bio-based conductive ink can therefore produce environmentally friendly and sustainable wood-based electronic products for potential applications such as flexible circuits and sensors. Indeed, the development of flexible and/or transparent electrical devices fabricated from natural wood using environmentally friendly design methods is expected to open up new possibilities in biomedical devices, smart electronics, and other fields. Considering the similarity between the structure and composition of bamboo and wood, the use of bamboo in such applications is not only feasible but could also improve sustainability.

## 6. Conclusions and Outlook

This review summarized the recent progress in TB, including different methods for preparing TB with improved optical, mechanical, and thermal conductivity properties; the realization of TB functionalization in energy-efficient building components; and discussions of potential TB use in decorative applications and optoelectronic devices.

The delignification process used to prepare TB will affect its optical and mechanical properties. Previous experimental results indicate that TB prepared by lignin modification has better properties than the TB prepared using other methods. However, lignin modification requires more chemical solvents than most other methods. Thus, several sustainability issues related to delignification technology optimization should be addressed in future research. For example, environmentally friendly “green” chemistry approaches are necessary to minimize the use of solvents, reduce reaction time and waste streams in the preparation of TB, and develop TB-specific functionalization methodologies. Proper resin selection is also essential to realizing the desired TB properties. However, as there have been few attempts to prepare TB using different resin types, different resins and resin modifications should be studied to increase the functionality of TB.Scaling up the production of TB is expected to be challenging, although some strategies targeting TB have been explored with promising results. Furthermore, the results reported to date have focused on small and thin TB samples. As an increase in thickness causes the light to travel longer and decay inside the TB, increasing thickness while maintaining sufficient transparency represents a significant challenge to the successful implementation of this technology at the industrial scale. Various technical optimization approaches are therefore worthy of investigation, such as the effect of parallel lamination and cross lamination on TB performance.The potential functionalizations of TB include its development as a: (i) thermal insulation material, (ii) decorative material, (iii) conductive material, and (iv) or magnetic material. To date, TB has typically been singly functionalized for a specific scenario. The versatility of TB should therefore be further evaluated, such as the simultaneous combination of thermal insulation and magnetic functions to address a variety of situations using the same material. The functional utilization of bamboo is far less than that of wood at present, so there remains a wide range of strategies to be explored to develop novel functionalized bamboo materials.The functionalization of TB enables its application to diverse fields such as: (i) energy-saving windows, (ii) decorative applications, and (iii) optoelectronic devices. Several applications targeting TB have been explored with promising results. As the research community pays increasing attention to sustainable development, it is likely that functional TB technology will continue to develop over the next several years.

It is a pleasant process to imagine the future integrated application of TB (Figure 9). TB with good optical and mechanical properties can find potential applications as windows and ceilings in a house or museum, which can replace the glass. The TB-based energy-saving window can improve energy efficiency due to its great thermal insulation properties. Simultaneously, this kind of window can scatter light, which makes the light distribution more uniform compared with normal glass. Similarly, the furniture and household, such as lampshades, desktops, and byobu made of TB, can soften the light and add a calm and tranquil scenery to the room. Different transparency and haze produced after the superposition of TB can form different visual effects. The bamboo-based furniture and household design reflect elements of Chinese traditional culture. In the design of optoelectronic devices, such as wearable sensors, while giving full play to the advantages of TB, more attention should be paid to combine theoretical calculation with theoretical simulation to realize the rapid development of modern bamboo science.

## Figures and Tables

**Figure 2 polymers-14-03234-f002:**
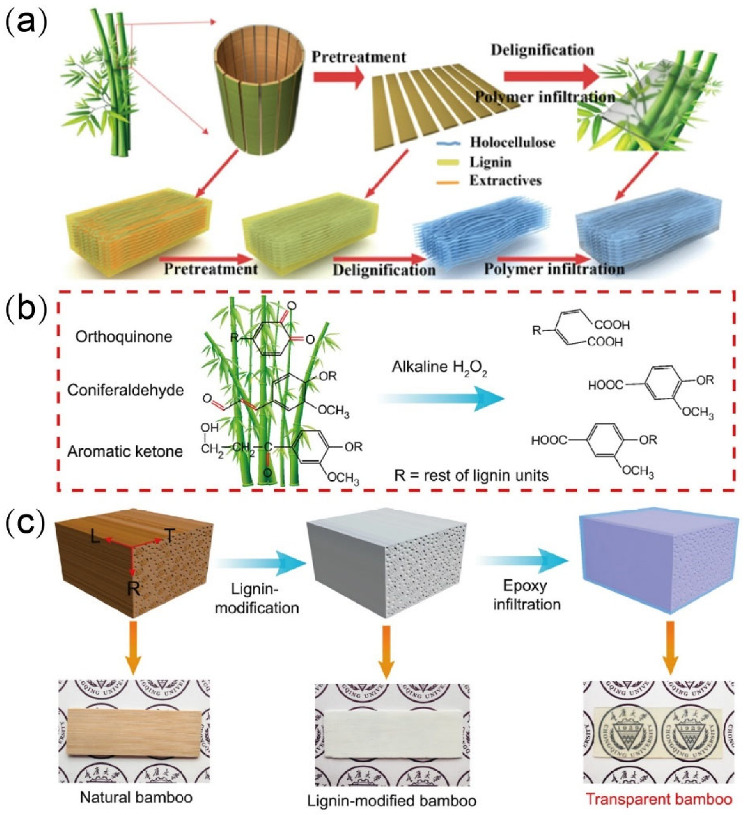
Preparation of transparent bamboo (TB) using the (**a**) acid delignification (including pretreatment) [23], (**b**,**c**) lignin modification [35] methods. ((**a**) Reproduced with permission from Wang et al. [23]. Copyright 2021 ACS Publications; (**b**,**c**) Reproduced with permission from Wang et al. [35]. Copyright 2022 Elsevier).

**Figure 3 polymers-14-03234-f003:**
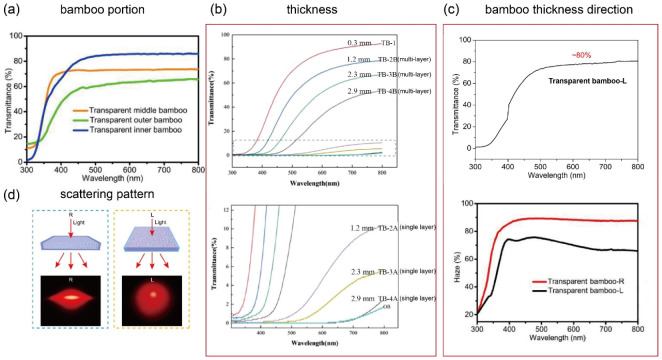
Transmittance and haze data for TB according to (**a**) bamboo fiber location [35], (**b**) bamboo thickness [24], and (**c**) bamboo thickness direction [35], as well as the (**d**) scattering pattern of TB according to bamboo thickness direction (anisotropy effect) [35]. ((**a**,**c**,**d**) Adapted from Wang et al. [35]; (**b**) Adapted from Wu et al. [24]).

**Figure 4 polymers-14-03234-f004:**
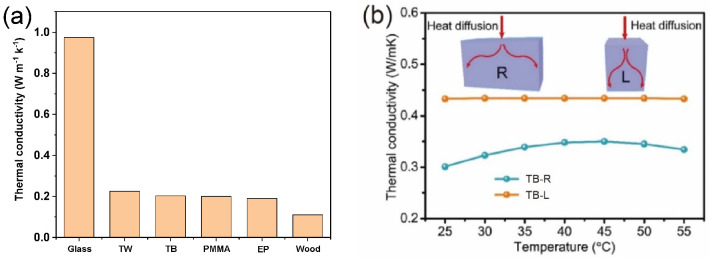
Thermal conductivity of TB: (**a**) comparison of the thermal conductivities of common glass, transparent wood (TW), TB, polymethyl methacrylate (PMMA), epoxy polymer (EP), and wood [23,52,53]; (**b**) thermal conductivities of TB in the longitudinal (L) and radial (R) directions [35]. ((**b**) adapted from Wang et al. [35]).

**Figure 5 polymers-14-03234-f005:**
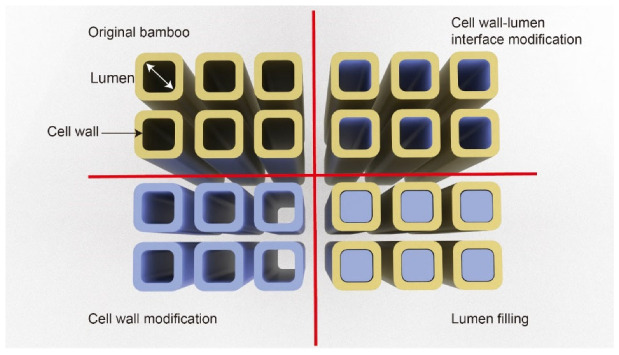
A selection of structural modification strategies to improve TB functionality (inspired by Burgert et al. [62]).

**Figure 7 polymers-14-03234-f007:**
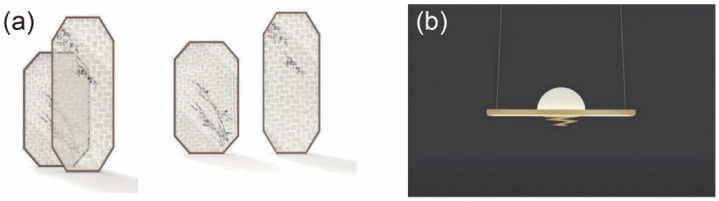
Decorative applications of TB: (**a**) woven composite screens; (**b**) sunset chandelier [76]. ((**a**,**b**) Reproduced with permission from ref. [76]. Copyright 2021 Editorial Department of *Journal of furniture.*).

**Figure 8 polymers-14-03234-f008:**
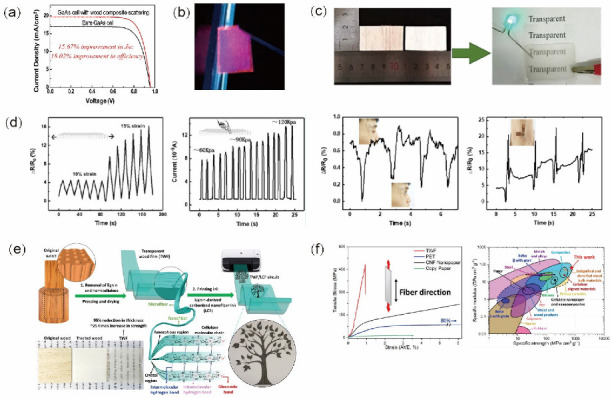
Application of TWC materials in optoelectronics: (**a**) current density versus voltage characteristics for a bare GaAs cell (black) and a GaAs cell with light management wood coating (red) [78]; (**b**) TW glowing under ultraviolet (UV) light [81]; (**c**) digital photograph of transparent conductive wood (TCW) with LED lighting [83]; (**d**) TCW used a strain/touch sensor to monitor human activities [83]; (**e**) processing of transparent wood film (TWF) for application in flexible electronics [84]; (**f**) mechanical properties of TWF compared with various other materials [84]. ((**a**) Adapted from Zhu et al. [78]; (**b**) Adapted from Gan et al. [81]; (**c**,**f**) Adapted from Fu et al. [84]; (**d**) Adapted from Wang et al. [83]).

**Figure 9 polymers-14-03234-f009:**
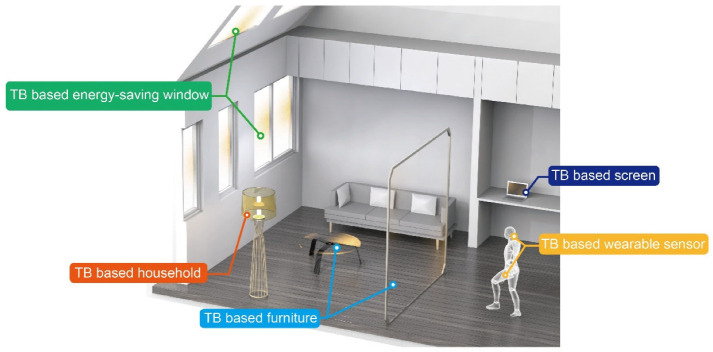
Design of a future integrated application of TB.

**Table 1 polymers-14-03234-t001:** Summary of TB preparations and properties.

Ref.	Template Preparation Method; Temperature; Time	Polymer	Variable	Optical Property			Mechanical Property	
				t (mm)	Tr (%)	haze (%)	l × w × t (mm^3^)	TS (MPa)
Wu et al. [25]	Delignification: NaClO_2_; 80–90 °C; 2–4 h	Epoxy (E51) (RI = approximately 1.5)	inner	1.1	approximately 12	—	3 × 4.4 × 1.1	35.31
			outer	1.8	approximately 2	—	3 × 7.8 × 1.8	82.18
Wang et al. [23]	Delignification: preconditioned in NaOH + 10 h; NaClO_2_; 85 °C; 3 h	Two-part epoxy resin (Clearcast 7000) (RI = approximately 1.5)	—	1	approximately 80	80	165 × 13 × 1	92
			—	1.5	approximately 75	80	165 × 13 × 1.5	—
Wu et al. [24]	Delignification: NaClO_2_; 80–90 °C; 2–3 h	Epoxy (E51) (RI = 1.52)	—	0.3	92.4	43.5	40 × 20 × 0.3	47.1
			multi-layer (3 layers)	1.2	78.6	70	40 × 20 × 1.2	61.89
			multi-layer (5 layers)	2.3	67.1	70.55	40 × 20 × 2.3	approximately 60
			multi-layer (7 layers)	2.9	23.7	82.95	40 × 20 × 2.9	approximately 60
			single layer	1.2	10.4	97.02	40 × 20 × 1.2	61.15
			single layer	2.3	5.5	~100	40 × 20 × 2.3	approximately 30
			single layer	2.9	1.7	~100	40 × 20 × 2.9	approximately 10
Wang et al. [35]	Lignin modification: lignin-modification solution; 70 °C; until become white	Epoxy (E51) (RI = approximately 1.5)	inner	1.5	87	—	—	—
			middle	1.5	74	—	—	—
			outer	1.5	66	—	—	—
			radial	1.5	—	90	20 × 8 × 1.5	30
			longitudinal	1.5	80	70	50 × 10 × 1.5	118

**Table 2 polymers-14-03234-t002:** Summary of mechanical and physical properties of natural plant fibres.

Ref.	Type of Fibre	Density (g/cm^3^)	Tensile Strength (MPa)	Young’s Modulus (MPa)
Abdul Khalil et al. [42]	Moso bamboo (*Phyllostachys pubescens*)	1.2–1.5	500–575	27–40
Liu et al. [3]	Oil palm	0.7–1.6	248	3.2
Liu et al. [3]	Pineapple	0.8–1.6	1.44	34.5–82.5
Cai et al. [43]	Abaca	1.5	717	18.6
Vijaya Ramnath et al. [44]	Jute	1.3–1.49	393–800	13–26.5
Ramesh et al. [45]	Sisal	1.41	350–370	12.8
Mohamad et al. [46]	Kenaf	1.2	282.6	7.13
Asim et al. [47]	Coconut	1.2	140–225	3–5

## Data Availability

Not applicable.

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
