# Peer review of "Optically Transparent Bamboo: Preparation, Properties, and Applications"

_polymers, 2022, doi:10.3390/polym14163234_

Round 1

Reviewer 1 Report

In this contribution, the authors give a thorough review of recent progress in epoxy-impregnated transparent bamboo (TB). Various preparation methods of TB are elaborated, and then the optical transmittance, mechanical performance and thermal conductivity in the radial/longitudinal directions are compared. Finally, the authors introduce the functionalization, applications and outlooks of TB. This review is informative and well organized. Besides, it is inspiring and beneficial for the readership of Polymers. Therefore, I would highly recommend its publication if the following comments/questions are addressed.  

1. In Line 257, the authors mention “the transmittance of TB generally increases with increasing fiber volume fraction”. However, the transmittance of TB should decrease with increasing fiber volume fraction, because bamboo fibers hinder the infiltration of lignin-modifying chemicals and result in more chromospheres. Figure 3a also shows a higher transmittance for inner layers, lower fiber volume fractions, than outer layers, higher fiber volume fractions. 

2. In Table 1, the work by Wu et al. shows the transmittance of approximately 2% for inner TB and 12% for outer TB. This seems to be opposite to Wu’s original results, where the transmittance is above 10% for TIB (transparent inner bamboo), and 2% for TOB (transparent outer bamboo).

3. The authors have a positive outlook on transparent bamboo for various applications, such as energy-saving windows and TB-based screens. The limited diameter and thickness of bamboo, as well as nodes and sulcus along the culm, however, restrict the dimension of uniform single layer TB. Can the authors also elaborate their thoughts on how to overcome this limitation?

4. Following the previous comment, if two pieces of birefringent TB are overlaid perpendicularly (cross lamination in Line 521), will the transmittance be different from those overlaid in parallel? 

5. Three kinds of highlighted woods for transparent TB (transparent wood) are listed in Line 73 because of their low densities. Are bamboos, among the 1500 species (Line 33), also featured with examples of preferred species due to low density or low fiber volume fractions?

Author Response

To Reviewer #1,

Dear professor,

We deeply appreciate the valuable suggestion about the manuscript (MS). Indeed, these comments are very useful for us to further improve the MS. Now we complete the revision of this work. We do hope you think this new version of the MS is satisfactory for publication.

There are 5 items in the comment in total. With high respect to you, we would be delighted to answer your point to point as follows:

Reviewers' comments:

Reviewer #1: In this contribution, the authors give a thorough review of recent progress in epoxy-impregnated transparent bamboo (TB). Various preparation methods of TB are elaborated, and then the optical transmittance, mechanical performance and thermal conductivity in the radial/longitudinal directions are compared. Finally, the authors introduce the functionalization, applications and outlooks of TB. This review is informative and well organized. Besides, it is inspiring and beneficial for the readership of Polymers. Therefore, I would highly recommend its publication if the following comments/questions are addressed.

  1. In Line 257, the authors mention “the transmittance of TB generally increases with increasing fiber volume fraction”. However, the transmittance of TB should decrease with increasing fiber volume fraction, because bamboo fibers hinder the infiltration of lignin-modifying chemicals and result in more chromospheres. Figure 3a also shows a higher transmittance for inner layers, lower fiber volume fractions, than outer layers, higher fiber volume fractions. 

Response: Thanks for the comments. We are sorry for your inconvenience caused by our negligence. We corrected and added the description in line 257 and 260. Please check it.

  1. In Table 1, the work by Wu et al.shows the transmittance of approximately 2% for inner TB and 12% for outer TB. This seems to be opposite to Wu’s original results, where the transmittance is above 10% for TIB (transparent inner bamboo), and 2% for TOB (transparent outer bamboo).

Response: Thanks for the comments. We feel really sorry for our carelessness. We corrected the data in table 2 according to your comment. Please check it.

  1. The authors have a positive outlook on transparent bamboo for various applications, such as energy-saving windows and TB-based screens. The limited diameter and thickness of bamboo, as well as nodes and sulcus along the culm, however, restrict the dimension of uniform single layer TB. Can the authors also elaborate their thoughts on how to overcome this limitation?

Response: Thanks for the nice comment on our article. Bamboo often overcame its size limit by breaking it down into different units and recombining them like particleboard [1], plywood [2,3], and other bamboo-layered structural elements in engineering. For example, bamboo veneer was obtained by rotary cutting. To overcome the limitation of original thickness, bamboo veneers were laminated in different ways to make bamboo-based panels. The bamboo nodes and sulcus on bamboo size limitation, we can make a reasonable selection of the bamboo parts and avoid the bamboo nodes. In addition, we can also keep the bamboo nodes properly. As one of the most important characteristic elements of bamboo, bamboo node is of great significance to the expression of Chinese culture and aesthetic value in the field of decorative aesthetics.

  1. Following the previous comment, if two pieces of birefringent TB are overlaid perpendicularly (cross lamination in Line 521), will the transmittance be different from those overlaid in parallel? 

Response: Thanks for the comments. As you said, the laminating method also has an effect on the transmittance [4]. The transmittance of parallel lamination is often higher than cross lamination. In addition, in comparison with parallel lamination, cross lamination changed the relatively single propagation path of the light beam and made the light more uniform.

  1. Three kinds of highlighted woods for transparent TB (transparent wood) are listed in Line 73 because of their low densities. Are bamboos, among the 1500 species (Line 33), also featured with examples of preferred species due to low density or low fiber volume fractions?

Response: Generally speaking, the density of most bamboo species was higher than that of wood. Compared to wood, bamboo had a gradient fiber structure. Inner layer of bamboo culm with low fiber volume fraction was more suitable for achieving high transmittance [5]. For outer layer with high volume fraction, the final TB obtained by delignification and resin impregnation had a more obvious and detailed texture. This was conducive to the application in the field of decorative aesthetics.

  1. Sarmin, S.N.; Zakaria, S.A.K.Y.; Kasim, J.; Shafie, A. Influence of Resin Content and Density on Thickness Swelling of Three-Layered Hybrid Particleboard Composed of Sawdust and Acacia Mangium. BioResources 2013, 8, 4864–4872, doi:10.15376/biores.8.4.4864-4872.
  2. Kallakas, H.; Rohumaa, A.; Vahermets, H.; Kers, J. Effect of Different Hardwood Species and Lay-up Schemes on the Mechanical Properties of Plywood. Forests 2020, 11, 1–13, doi:10.3390/f11060649.
  3. Fu, Q.; Yan, M.; Jungstedt, E.; Yang, X.; Li, Y.; Berglund, L.A. Transparent Plywood as a Load-Bearing and Luminescent Biocomposite. Compos. Sci. Technol. 2018, 164, 296–303, doi:10.1016/j.compscitech.2018.06.001.
  4. Wang, J.; Wang, Y.; Wu, Y.; Zhao, W. A Multilayer Transparent Bamboo with Good Optical Properties and UV Shielding Prepared by Different Lamination Methods. ACS Sustain. Chem. Eng. 2022, 10, 6106–6116, doi:10.1021/acssuschemeng.2c01719.
  5. Wu, Y.; Wang, Y.; Yang, F.; Wang, J.; Wang, X. Study on the Properties of Transparent Bamboo Prepared by Epoxy Resin Impregnation. Polymers (Basel). 2020, 12, doi:10.3390/POLYM12040863.

 Again, the authors wish to thank the reviewers for their thoughtful comments and their work on our manuscript.

With best regards

Sincerely yours,

Fei Rao*, [email protected]

Zhang Weizhong, [email protected]

School of Art and Design, Zhejiang Sci-Tech University, No. 2 street 928, Hangzhou 310018, China

Reviewer 2 Report

Dear Author,

The manuscript is interesting especially related to transparency which can be helpful for a broader impact on the community. Here are my comments:

(1) Figure 3 is difficult to read. The legend should be made bigger and readable.

(2) Mechanical data, as well as graphs, should be shown for the readers to understand how these fibers are and how mechanically strong these bamboo fibers.

(3) Author failed to explain the origin of transparency and how chromophores helps the transparency

Author Response

To Reviewer #2,

Dear professor,

Here we must express our deep gratitude firstly to you for spending time to evaluate the publication of this manuscript (MS) and the visionary comments, which are of great help to guide our follow-up research work.

Reviewer #2: The manuscript is interesting especially related to transparency which can be helpful for a broader impact on the community. Here are my comments:

  1. Figure 3 is difficult to read. The legend should be made bigger and readable.

Response: Thanks for the nice comment. We are sorry for your inconvenience caused by our negligence. Based on the suggestions, we have made an extensive modification on the revised manuscript. Detailed revision was shown as follows:

  1. Enlarge the legend as a whole.
  2. Adjust font size

Please check it.

  1. Mechanical data, as well as graphs, should be shown for the readers to understand how these fibers are and how mechanically strong these bamboo fibers.

Response: Thanks for the comment. We added a table 1 including mechanical and physical properties of natural plant fibres. Please check it.

  1. Author failed to explain the origin of transparency and how chromophores help the transparency.

Response: Thanks for your comment. Bamboo is opaque because of its optically heterogeneous nature, a result of its microscale porous structure, different chemical components with different refractive index (RIs) in the cell walls, and contents of strongly light-absorbing chemical entities such as lignin.

Generally, to make bamboo transparent, light absorption by chemical entities and light scattering at the air/cell wall interfaces and inside the cell walls must be reduced or eliminated. So we used delignification to remove the lignin and epoxy resin impregnation to match the refractive index between the cellulose in the cell wall and the air.

Again, the authors wish to thank the reviewers for their thoughtful comments and their work on our manuscript.

With best regards

Sincerely yours,

Fei Rao*, [email protected]

Zhang Weizhong, [email protected]

School of Art and Design, Zhejiang Sci-Tech University, No. 2 street 928, Hangzhou 310018, China
